# Resolving the prevalence of somatic transposition in *Drosophila*

**Christoph D Treiber\*, Scott Waddell\***

Centre for Neural Circuits and Behaviour, The University of Oxford, Oxford, United Kingdom

**Abstract** Somatic transposition in mammals and insects could increase cellular diversity and neural mobilization has been implicated in age-dependent decline. To understand the impact of transposition in somatic cells it is essential to reliably measure the frequency and map locations of new insertions. Here we identified thousands of putative somatic transposon insertions in neurons from individual *Drosophila melanogaster* using whole-genome sequencing. However, the number of de novo insertions did not correlate with transposon expression or fly age. Analysing our data with exons as 'immobile genetic elements' revealed a similar frequency of unexpected exon translocations. A new sequencing strategy that recovers transposon: chromosome junction information revealed most putative de novo transposon and exon insertions likely result from unavoidable chimeric artefacts. Reanalysis of other published data suggests similar artefacts are often mistaken for genuine somatic transposition. We conclude that somatic transposition is less prevalent in *Drosophila* than previously envisaged.

**\*For correspondence:** christoph. d.treiber@gmail.com (CDT); scott. waddell@cncb.ox.ac.uk (SW)

**Competing interests:** The authors declare that no competing interests exist.

## Introduction

Mobile genetic elements, or transposons, constitute up to 70% of eukaryotic genomes (*Adams et al., 2000*; *Britten and Kohne, 1968*; *Lander et al., 2001*; *Waterston et al., 2002*) and are believed to have contributed variation, upon which evolution has acted (*Kazazian, 2004*; *Levin and Moran, 2011*). Several studies have also suggested that transposition introduces non-heritable genetic heterogeneity in somatic cells, including neurons in the brain (*Muotri et al., 2005*; *Coufal et al., 2009*; *Baillie et al., 2011*; *Evrony et al., 2012*; *Kazazian, 2011*; *Perrat et al., 2013*). Stochastic transposition into neural genes could alter the functional range of a particular group of neurons, in addition to contributing towards behavioral individuality within a species. Ongoing mobilization can also be problematic. Somatic transposition has been implicated in age-dependent neural and cognitive decline (*Bundo et al., 2014*; *Krug et al., 2017*; *Li et al., 2013*) and tumorigenesis in other tissues (*Shukla et al., 2013*; *Solyom and Kazazian, 2012*). However, the prevalence of rare somatic transposition is debated due to difficulties in mapping genuine events using whole-genome DNA sequencing (*Baillie et al., 2011*; *Evrony et al., 2012*, *Evrony et al., 2016*; *Upton et al., 2015*).

Individual somatic transposon insertions are difficult to detect, because they occur in single DNA molecules. To identify de novo insertion sites in the genome these molecules need to be extracted from tissue, purified, amplified, sequenced and analyzed. Unfortunately, each of these steps is associated with considerable pitfalls. During genomic DNA (gDNA) extraction, nuclei of interest need to be separated from the rest of the biological material. Pooling cells at this stage increases the total yield of DNA but it lowers the power to detect unique insertions in individual cells. Although single-cell based approaches provide the highest theoretical sensitivity to detect rare insertions, the percentage of the genome that is covered for each cell, and the total number of cells that can be tested with this approach, is limited (*Gawad et al., 2016*). Prior work has suggested that transposons are

expressed and mobile in the brains of *Drosophila melanogaster* fruit flies, and in particular in clonally related sub-populations of mushroom body neurons, called αβ-Kenyon Cells (αβ-KCs) (*Li et al., 2013*; *Perrat et al., 2013*). Since each fly has ~1800 αβ-KCs (*Aso et al., 2014*), these cells are ideal to study how transposition might create cellular diversity. Moreover, the αβ-KCs have a defined role in the retrieval of consolidated memory (*Krashes and Waddell, 2008*; *Krashes et al., 2007*), making it possible to assess the impact that transposition might have on cognitive function.

We developed a method to extract and sequence gDNA with high genome-wide coverage from most of the ~1800 αβ-KCs from an individual fly, and from cells taken from the rest of the brain of the same fly. Using this approach, we analyzed rates of de novo somatic transposition using a standard detection algorithm (*Zhuang et al., 2014*). Transposon sequences did not accumulate with age. In addition, although we found some transposons are more highly expressed in αβ-KCs, when compared to the rest of the brain, their upregulation did not lead to a measurably larger number of putative insertions in αβ-KC gDNA. Reanalysis with a novel simulated set of 'immobile genetic elements', rather than transposons, revealed that our Whole Genome Sequencing (WGS) data and that from other published work (*Khurana et al., 2011*) contain unexpected sequences indicative of exon translocations. A new analysis pipeline that retrieves the precise nucleotide sequence around potential DNA breakpoints revealed that these translocations and the majority of putative de novo transposon insertions result from chimeric artefacts formed during WGS library preparation. Although previous studies of somatic transposition raised the issue of chimeric DNA sequences influencing the reliability of mapping transposon insertions (e.g. *Evrony et al., 2016*), we here provide evidence that chimera are prevalent in WGS data prepared from a eukaryote. Moreover, we present a new approach to assess the abundance of these chimera in any WGS data set. Taken together our findings highlight a fundamental flaw in current approaches to detect and evaluate rare somatic transposon insertions and they challenge the prior hypotheses that transposition plays a major role in the generation of cellular diversity and in age-dependent neuronal decline.

## Results

### Extraction of pure αβ neuron populations from fly brains

Previous studies suggested that some transposons are expressed in the fly brain, and perhaps at higher levels in αβ-KCs of the mushroom body (*Li et al., 2013*; *Perrat et al., 2013*). In our earlier study αβ-KCs were labeled with intersectional genetics based on site-directed DNA recombination and these KC populations from groups of flies were purified with Fluorescence Activated Cell Sorting (FACS) (*Perrat et al., 2013*). mRNA was then amplified with exome-wide in vitro transcription and differences in mRNA expression levels were assessed with a microarray strategy. The stochastic cell labeling inherent to this approach makes it difficult to compare cells from the same animals because some desired cells are not labeled and so contaminate the other cell sample. In this study we instead used a split-GAL4 fly line, MB008b-GAL4, to specifically label a similar number of αβ-KCs in every fly with UAS-mCherry (*Figure 1a,b*) (*Aso et al., 2014*; *Luan et al., 2006*) and again used FACS to separate fluorescent αβ-KCs from groups of flies from the rest of the cells in the same brains (*Figure 1c*). We verified the purity of the FACS populations by measuring the expression level of mCherry in both fractions. RNA was extracted, target RNA was amplified following multiplexed pre-amplification, and expression levels were assessed using real-time quantitative PCR (RT-qPCR). This analysis showed that mCherry is upregulated by approximately 1000-fold in the purified αβ-KCs (*Figure 1d*). We also tested levels of mRNA from the *FasII* gene, which can be readily detected in αβ-KCs with immunostaining (*Crittenden et al., 1998*). As expected *FasII* was upregulated by 5-fold in αβ-KCs. Our FACS approach therefore efficiently separates labeled αβ-KCs from the rest of the cells in the brain.

### Analysis of neural transposition in individual fruit flies

Somatic transposon insertions were previously assessed following gDNA extraction from neurons purified from groups of flies (*Perrat et al., 2013*). However, the approach of pooling gDNA from multiple flies lacks the power to reliably detect rare somatic insertion events at the sequencing depths that can currently be achieved. Since there are approximately 1800 labeled αβ-KCs in each fly brain and each genetic locus occurs twice in each cell, an experimental approach with groups of

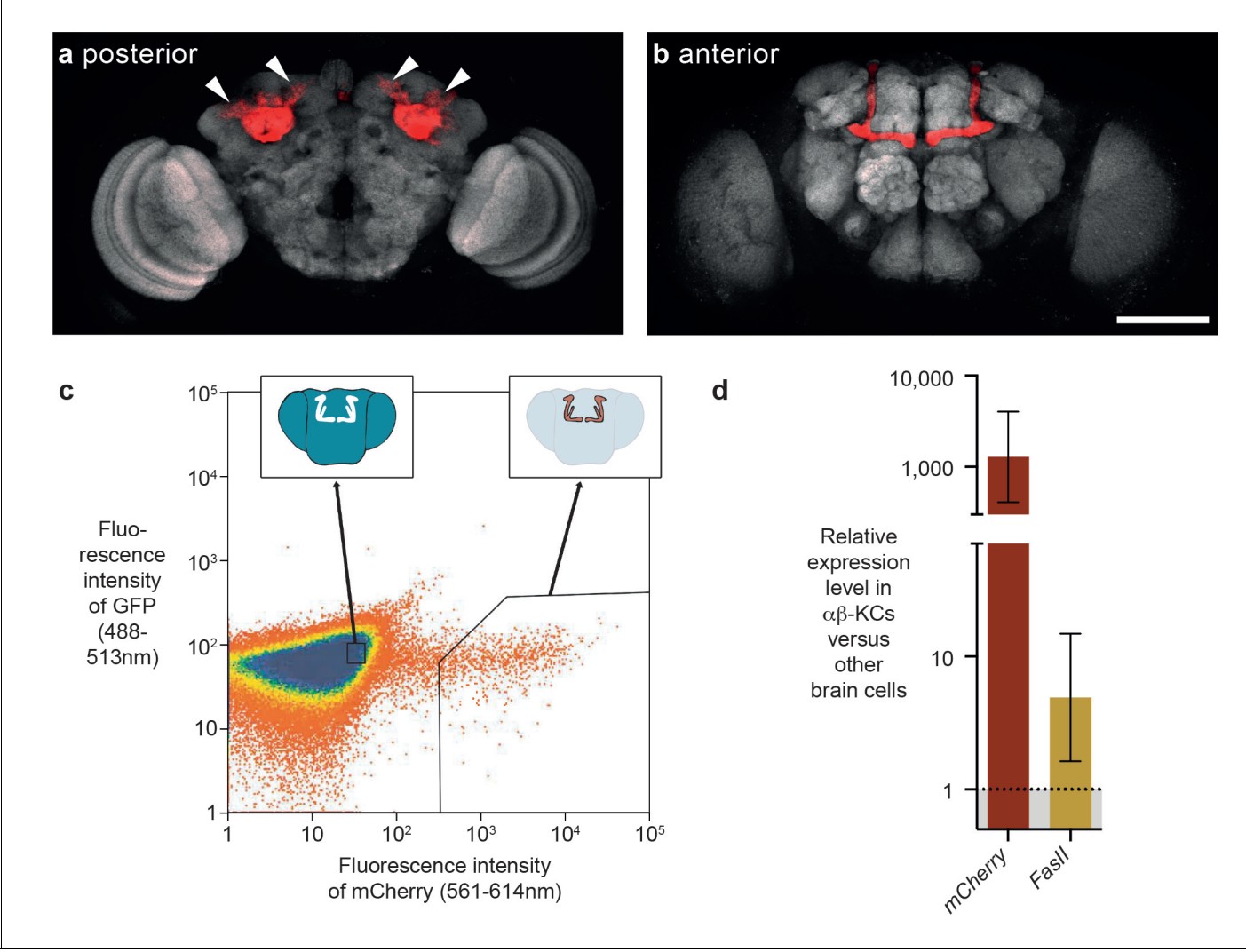

**Figure 1.** FACS strategy to extract αβ-Kenyon cells (αβ-KCs) from fly brains. (**a**) Posterior projection view of a confocal microscope stack showing αβ-KC somata (arrow heads) in a female *Mb008b-GAL4; UAS-mCherry* fly brain. The general neuropil is stained with the anti-bruchpilot antibody nc82 (white) and red indicates αβ-KC mCherry expression. (**b**) Anterior projection from the same brain as in (**a**) showing axons of the mushroom body lobes of αβ-KCs in either brain hemisphere, which form distinct bilaterally symmetrical L-shaped projections into vertical and horizontal lobes. This lobe structure will be used in all schematic representations of the brain in the rest of this manuscript to indicate the source of each gDNA sample. Scale bar 100 μm. (**c**) Example plot from a FACS run of a single fly brain to illustrate the selection strategy used to sort mCherry-positive αβ-KCs from mCherry-negative cells in dissociated brain tissue. Sorting gates were hand-drawn, with the aim of selecting a pure proportion of mCherry αβ-KCs (right inset shows a simplified fly brain highlighting αβ-KCs) and a second population of similar size, from the mCherry negative cells in the rest of the brain (left inset). Single cells are represented as red points and areas of high-density are colored blue. The same gates were used for all samples in this study. (**d**) mCherry and *FasII* expression is elevated in mCherry positive cell fractions. Graph shows relative expression levels in FAC-sorted αβ-KCs as compared to that in unlabeled cells. Error bars denote standard error of the mean (SEM).

30 flies generates a sample containing 108,000 copies of every locus. At a sequencing depth of 25x, in theory only 1 in 4320 events that occurred in one locus of one cell can be detected. Therefore, the chance of finding a rare de novo event is extremely low. In addition, at this discovery rate, bona fide new transposition events are indistinguishable from low frequency polymorphic germline transposon insertions present in the population of flies. Given these shortcomings, we developed a protocol to analyze the genomes of αβ-KCs and of cells taken from the rest of the brain (from here on referred to as 'other brain cells') from the same individual flies. This allows comparison of gDNA from two populations of cells which all originate from the same oocyte. Hence, every difference in

the sequence between these two groups should arise from somatic mutations that occurred during the development or adult life of that individual fly. We dissociated neurons from six individual fly brains and again purified the cells using FACS. However, this time gDNA was extracted and amplified with multiple-displacement amplification (MDA) (*Dean et al., 2001*). Finally, we created DNA libraries from the fragmented gDNA and subjected that material to WGS. Sequencing data obtained from each individual fly was aligned to the *Drosophila* reference genome and all samples exhibited an even distribution of reads across the fly genome (*Figure 2a*).

Almost all current strategies to detect somatic transposon insertions are based on the analysis of discordant read pairs in paired-end sequencing data, where one read maps onto a mobile element, and the other is used to determine the location of the insertion in the genome (e.g. TEBreak [*Carreira et al., 2016*], T-Lex [*Fiston-Lavier et al., 2011*], RetroSeq [*Keane et al., 2013*], RelocaTE [*Robb et al., 2013*], TEMP [*Zhuang et al., 2014*]). Since we used TEMP in our previous study (*Perrat et al., 2013*), we used it again here. We first validated our new sequencing data and the utility of TEMP by mapping transposon insertions in each of our DNA samples. Using this analysis we identified 163 new transposon insertions that are present in our MB008b-GAL4GAL4; UAS-mCherry flies but that are absent in the reference genome. Importantly, these same 163 insertions were common to all 12 samples prepared from six individual flies demonstrating that they represent germline insertions that are unique to our fly strain and that they have not been mapped before (*Supplementary file 1*).

TEMP also revealed on average 10,881 putative $\alpha\beta$-KCs specific transposition events per fly (N = 6, SD = 1960, see *Table 1*) that were absent in the gDNA sequenced from the rest of the brain cells from the same individual fly. A similar number of apparent insertions were found exclusively in the other brain cells and not in $\alpha\beta$-KCs from the same animal. Surprisingly, the positions of these hits revealed a recurring and very particular pattern where putative de novo somatic insertions appeared to cluster approximately 10 kb up- and down-stream of pre-existing germline locations of transposons, a phenomenon that we termed 'hovering' (*Figure 2b*). In the example shown in *Figure 2b*, a germline insertion of the *Doc* transposon is inserted in an intron of the *turtle* gene (*Al-Anzi and Wyman, 2009*). TEMP identified additional copies of *Doc* hovering around the germline insertion, including some inside *turtle* exons. An exonic insertion would interrupt the amino acid sequence of the *turtle* gene product in a subset of $\alpha\beta$-KCs and could potentially alter the neural wiring and thus functional diversity of $\alpha\beta$-KCs. The apparent phenomenon of transposon hovering was evident in both $\alpha\beta$-KC gDNA (*Figure 2b*, dark green traces) and that from cells in the rest of the brain (*Figure 2b*, light green traces). Finding that the total number of putative somatic insertions was similar in $\alpha\beta$-KCs and in the other brain cells (see *Table 1*), despite the reported elevated transposon activity in $\alpha\beta$-KCs[19], and that both types of samples exhibited the hovering phenomenon was unexpected and made us question the legitimacy of the putative de novo insertions.

## Assessing the rate of putative transposon insertions

Retrotransposition is replicative, which means that every new insertion should increase the number of copies of the respective retrotransposon in the host cell. Consequently, if retrotransposition is ongoing in somatic cells, the number of new insertions should accumulate over the life of a fly (*Li et al., 2013*). Three of our individually sequenced flies were 3 days old and the other three were 30 days old. We therefore analyzed the number of insertions of each transposon type identified in our TEMP analyses. We did not find a significant difference in the number of putative $\alpha\beta$-KC-specific insertions in gDNA samples prepared from young or old flies (*Figure 3a*). Since the linear amplification of gDNA in our sequencing approach should result in more sequencing reads for any transposon that has increased its copy number, we also counted the number of reads for each known transposon in gDNA from young and old flies. Again, no significant difference was apparent (*Figure 3b,c*).

It is widely believed that high level transposon expression translates to significant transposon mobility (*Li et al., 2013*). Hence it would seem reasonable to predict that more highly expressed transposons produce more somatic insertions. In contrast to LINE-1 retrotransposons in mammals, the DNA sequences of active fly transposons can easily be distinguished because there are many different families of elements (*Kaminker et al., 2002*). Therefore, we took advantage of their unique sequences to test whether high expression in $\alpha\beta$-KCs resulted in more insertions. We first determined the relative expression levels of 5 types of transposons in $\alpha\beta$-KCs extracted from the same

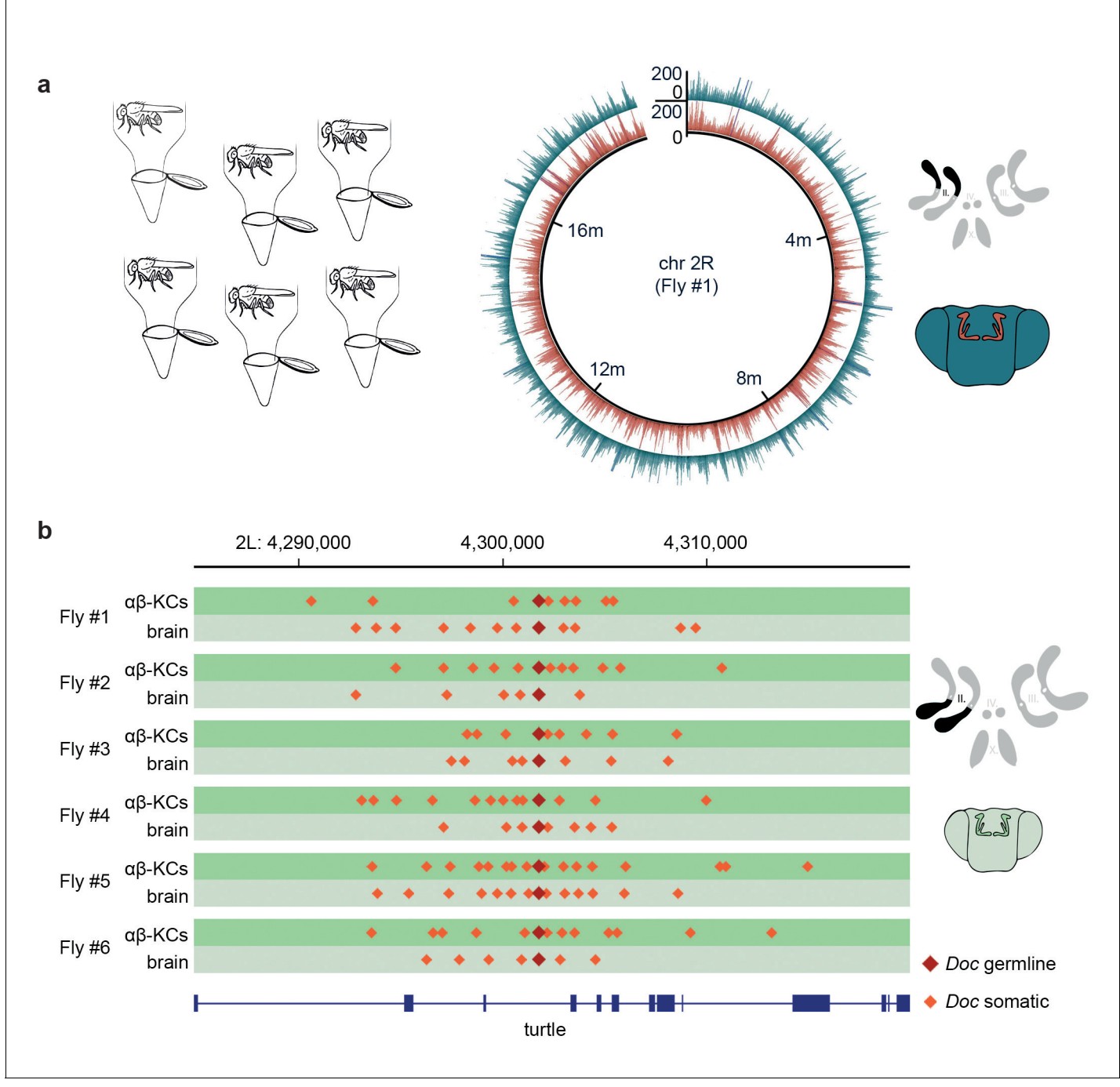

**Figure 2.** Single fly αβ-KC sequencing suggests transposon hovering. (**a**) Schematic of the experimental approach. Six individual flies were processed independently. The circular plot shows the gDNA sequencing coverage of mCherry positive αβ-KCs (red trace) and mCherry negative cells from the rest of the brain (blue trace), on chromosome 2R from one representative individual fly. The schematic (top right) depicts the 4 fruit fly chromosome pairs. Chromosome 2R, which is the source data in the circular plot, is highlighted in black. Schematic fly brain (bottom right) indicates the color scheme; αβ-KCs (red), the rest of the brain (blue). Sequencing read alignments on other regions of the gDNA exhibited a similar coverage (data not shown). (**b**) Plot of a representative example of a germline transposon insertion that was found on chromosome 2L in each of the 6 individual flies and that is absent in the *Drosophila melanogaster* reference genome (Release 5.57). Putative new insertions were found at loci, which were approximately 10 kb up- and downstream of the germline insertion site of the same transposon type. Dark red diamonds represent the germline insertion of the transposon *Doc*, which was found in each of the 12 samples (αβ-KCs and the rest of the brain), and light red diamonds represent putative somatic *Doc* insertions. The genomic location of the *turtle* gene is shown below in blue. Boxes indicate exons and lines intronic regions of the gene. Schematic fly brain represents the color code used for this panel.

*Figure 2 continued on next page*

*Figure 2 continued*

The following figure supplement is available for figure 2:

**Figure supplement 1.** Total number of putative non-reference somatic TE insertions from 12 samples with 1, 2, 3–9 and more than 10 diagnostic reads.

strain of flies used in this study, compared to expression in other brain cells. We again used FACS, followed by RNA extraction, target amplification and RT-qPCR (as described above). The LTR *Tabor* and LINE-like *Ivk* were upregulated by approximately 3-fold in αβ-KCs whereas the LTR retrotransposons *Gypsy* and *Blood* and the LINE-like *Doc3* were expressed at similar levels throughout the brain (*Figure 3d*). We next plotted these expression levels against the respective number of putative αβ-KC somatic insertions. No significant correlation was apparent, which suggests that our putative insertions did not result from insertion of more actively transcribed transposons.

The number of copies of each type of transposon in the *Drosophila* genome ranges from one to a few hundred (*Kaminker et al., 2002*). We therefore also quantified the number of sequences of each transposon family in the reference genome. The low copy number LINE-like *G7* only covers approximately 2000 nucleotides, while the high copy number LTR *roo* occupies more than 1,500,000 nucleotides in the reference genome. A comparison between the number of transposon nucleotides in the reference genome and the number of putative insertions of each transposon type in our αβ-KC sample gDNA revealed that these two values were highly correlated (Spearman test, p<0.0001) (*Figure 3e*). A linear regression of the log-transformed genome sequence copy numbers (normalized to read length and sequencing coverage) and the putative log-transformed number of insertions fitted the data well ($R^2 = 0.7166$). The slope of the fitted line was close to 1 (k = 0.9194 ± 0.04542), suggesting that the average number of detected putative new insertions for each transposon correlates approximately 1:1 with its abundance in the genome. This correlation is consistent with a scenario where the majority of identified putative de novo insertions actually result from random sampling of stretches of sequences across the *Drosophila* genome, which further questions the authenticity of the mapping of putative de novo insertions.

## 'Immobile genetic elements' reveal a high false discovery rate

Several of the hallmarks of a functional, specialized and biologically plausible regulation of transposition failed to emerge from our analysis. We did not observe more putative insertions in αβ-KCs than in other brain cells, no measurable increase in the number of insertions with age, no correlation of insertion number with expression level and the apparent insertion frequencies were consistent with random sampling of the genome. We therefore directly investigated whether the putative new insertions might result from experimental artefacts. We reasoned that if apparent new insertions arose from an inherent issue of our approach, any arbitrary stretch of DNA might appear to 'mobilize' when analyzed with TEMP. We tested this hypothesis by selecting a list of exon sequences from fly genes (see Methods) to create a set of 'immobile genetic elements', IGEs, based on the assumption that exons should not be mobile. The total number of nucleotides represented in the IGEs was the same as that in the transposon reference sequences. We also created a modified reference genome, DMsim, in which the IGE sequences were excised from their original location and were appended to the end of chromosome 2. For example, if the genome sequence normally consisted of region A, followed by region B and then C and we selected region B as an IGE, then the corresponding stretch of DNA sequence in DMsim would be A-C. Since the IGEs should be present at their original location in all flies, WGS data should contain DNA fragments that span the junctions of the IGE on both ends (A-B and B-C). When these sequencing reads are subsequently aligned to DMsim, the A-B and B-C spanning fragments map as discordant read pairs in this particular locus. These discordant read pairs are then screened for 'insertions', using the IGE sequence collection as a reference (rather than transposons), which should identify IGE B as a germline 'insertion' between regions A and C. Therefore, every genomic locus where we have removed an IGE in the DMsim reference genome should appear as a germline IGE insertion in every WGS data set. Importantly, the percentage of IGE insertions that are detected by TEMP in each individual WGS data set is a reliable indicator of the sensitivity to detect germline insertions in a particular gDNA sample. Moreover, finding any additional discordant IGE-genome read pairs indicates 'mobilization' of an IGE, which can be considered to be

**Table 1.** Summary of whole-genome sequencing data and TEMP results in this study. The number of artefactual IGE insertions that were detected in each sample are also shown. Note that the number of IGE insertions (column 'Putative IGE insertions') is a useful quality control metric to estimate the rate of chimera formed during amplification. Furthermore, the number of correctly identified IGEs (last column), in combination with the mean sequencing coverage, can be used to assess how equally distributed the read pairs are for each sample.

| Age | Sample number | Tissue sample | Read lengths | Total reads | % of reads mapped | Mean coverage | Range of insert sizes (of 90%) | Putative transposon insertions | Putative IGE insertions | Transposons only in αβ-KCs | IGEs only in αβ-KCs | Correctly identified IGEs (of 589) |
|---|---|---|---|---|---|---|---|---|---|---|---|---|
| YOUNG | 1 | αβ-KCs | 100 | 6E+07 | 96.96% | 39.43 | 434-500nt | 23301 | 2336 | 12827 | 1350 | 583 |
| | 2 | other brain cells | 100 | 6E+07 | 94.96% | 38.54 | 434-496nt | 22162 | 2222 | | | 583 |
| YOUNG | 3 | αβ-KCs | 100 | 6E+07 | 93.21% | 38.02 | 432-496nt | 22581 | 2322 | 12783 | 1179 | 583 |
| | 4 | other brain cells | 100 | 6E+07 | 96.01% | 39.21 | 434-504nt | 17224 | 2436 | | | 579 |
| YOUNG | 5 | αβ-KCs | 100 | 6E+07 | 96.57% | 39.13 | 438-508nt | 23974 | 2280 | 10977 | 1121 | 582 |
| | 6 | other brain cells | 100 | 6E+07 | 93.45% | 37.2 | 452-510nt | 24887 | 2234 | | | 582 |
| OLD | 7 | αβ-KCs | 100 | 6E+07 | 96.58% | 36.99 | 450-506nt | 17092 | 1794 | 8133 | 723 | 582 |
| | 8 | other brain cells | 100 | 6E+07 | 97.90% | 37.25 | 458-518nt | 16692 | 1777 | | | 582 |
| OLD | 9 | αβ-KCs | 100 | 6E+07 | 97.14% | 36.97 | 458-514nt | 18922 | 1998 | 8954 | 844 | 584 |
| | 10 | other brain cells | 100 | 6E+07 | 96.96% | 37.05 | 450-510nt | 19254 | 1950 | | | 581 |
| OLD | 11 | αβ-KCs | 100 | 6E+07 | 96.29% | 38.72 | 437-497nt | 24587 | 2181 | 11616 | 989 | 582 |
| | 12 | other brain cells | 100 | 6E+07 | 90.60% | 36.5 | 435-509nt | 21849 | 2241 | | | 582 |
| YOUNG | 13 | αβ-KCs | 250 | 7E+06 | 91.82% | 4.7 | 250nt | 1530 | 476 | 759 | 84 | 442 |
| | 14 | other brain cells | 250 | 8E+06 | 89.30% | 5.17 | 250nt | 1687 | 501 | | | 464 |
| YOUNG | 15 | αβ-KCs | 250 | 2E+07 | 89.68% | 10.45 | 250nt | 2498 | 625 | 1671 | 230 | 415 |
| | 16 | other brain cells | 250 | 9E+06 | 87.18% | 5.27 | 250nt | 1828 | 523 | | | 467 |
| YOUNG | 17 | αβ-KCs | 250 | 8E+06 | 78.12% | 6.41 | 250nt | 2189 | 212 | 1544 | 93 | 193 |
| | 18 | other brain cells | 250 | 1E+07 | 64.74% | 5.72 | 250nt | 1908 | 279 | | | 255 |
| OLD | 19 | αβ-KCs | 250 | 9E+06 | 89.78% | 5.6 | 250nt | 1732 | 470 | 1008 | 87 | 420 |
| | 20 | other brain cells | 250 | 9E+06 | 91.85% | 5.43 | 250nt | 1814 | 515 | | | 458 |

*Table 1 continued on next page*

*Table 1 continued*

| Age | Sample number | Tissue sample | Read lengths | Total reads | % of reads mapped | Mean coverage | Range of insert sizes (of 90%) | Putative transposon insertions | Putative IGE insertions | Transposons only in αβ-KCs | IGEs only in αβ-KCs | Correctly identified IGEs (of 589) |
|---|---|---|---|---|---|---|---|---|---|---|---|---|
| OLD | 21 | αβ-KCs | 250 | 8E +06 | 90.51% | 5 | 250nt | 1627 | 491 | 1017 | 136 | 445 |
| | 22 | other brain cells | 250 | 7E +06 | 90.26% | 5.01 | 250nt | 1575 | 458 | | | 396 |
| OLD | 23 | αβ-KCs | 250 | 8E +06 | 90.73% | 5.39 | 250nt | 1901 | 436 | 1204 | 124 | 388 |
| | 24 | other brain cells | 250 | 8E +06 | 84.21% | 4.83 | 250nt | 1661 | 427 | | | 373 |

a library artefact. Therefore, IGE analyses are an invaluable quality control metric for next generation sequencing data because the total number of additional IGE insertions is indicative of the false discovery rate for each gDNA sample.

Our analyses detected many read-pairs in all of our WGS samples that qualified as evidence for new IGE insertions (*Table 1*). The average frequency of these IGE insertions in αβ-KCs was 1034 (SD = 229.5, N = 6), a number lower than that for transposon sequences. However, when corrected for the lower copy numbers of IGEs (~30 fold), this value is comparable to that of putative transposon mobilizations detected by TEMP. This finding further supports the hypothesis that most putative insertions of transposons and IGEs actually represent randomly sampled genome fragments. It is also striking that apparent IGE mobilizations exhibited a similar hovering pattern around their original locations, as observed for transposons around their respective germline positions, suggesting this phenomenon is also artificial (*Figure 4a*).

To test whether sequencing artefacts, observable as IGE mobilization, are a general issue, we performed IGE experiments on WGS data from a published data set from another lab (*Khurana et al., 2011*). Female offspring from a cross of male flies carrying the *P*-element transposon (P) and female flies lacking the *P*-element (M) exhibit reduced fertility due to *P*-element mobilization, a phenomenon termed P-M hybrid dysgenesis (*Rubin et al., 1982*). P-M hybrid dysgenesis was reported to trigger the activation of resident transposons and 2–4 day old dysgenic ovaries were described to contain thousands of de novo transposon insertions when compared to their parental fly strains (*Khurana et al., 2011*). Moreover, the number of new insertions apparently further increased in 21 day old flies. When P-M dysgenic females were crossed to wild-type males, the number of new insertions detected in the progeny appeared to drop, a phenomenon that the authors ascribed to inheritance of appropriately targeted piRNA clusters.

We first repeated the TEMP analysis on the data published by *Khurana et al. (2011)* and found very similar rates of apparent transposon mobilization (*Figure 4b*, left panel). However, IGE experiments also detected considerable levels of IGE mobilization (*Figure 4b*, right panel) and most strikingly, the number of IGE 'mobilizations' was highest in oocytes of 21 day old dysgenic flies, when compared to oocytes of 4 day old flies, and lowest in the sample taken from flies that were backcrossed to the parental strain. The different levels of artefacts in each sample therefore follow the apparent observed differences in resident transposon mobilization. The authors also reported that the penetrance of a subset of somatic insertions increased with the age of flies. However, we found a comparable effect of the penetrance of IGE events in the respective samples (*Figure 4c*). Although IGE insertions should all have a penetrance of 1 in a perfect WGS data set, we only identified a subset of IGE insertions to have a penetrance of 1 in the *Khurana et al. (2011)* samples, while others diverge from this value. Lastly, similar to our observations with our new αβ-KCs data, there is a strong correlation between the number of putative transposon insertions identified in the oocyte samples and the abundance of each transposon type in the *Drosophila* genome, regardless of whether they are actively expressed or truncated and hence dysfunctional (Spearman test, p<0.0001) (*Kaminker et al., 2002*) (*Figure 4d*). Although *Khurana et al. (2011)* noted that there

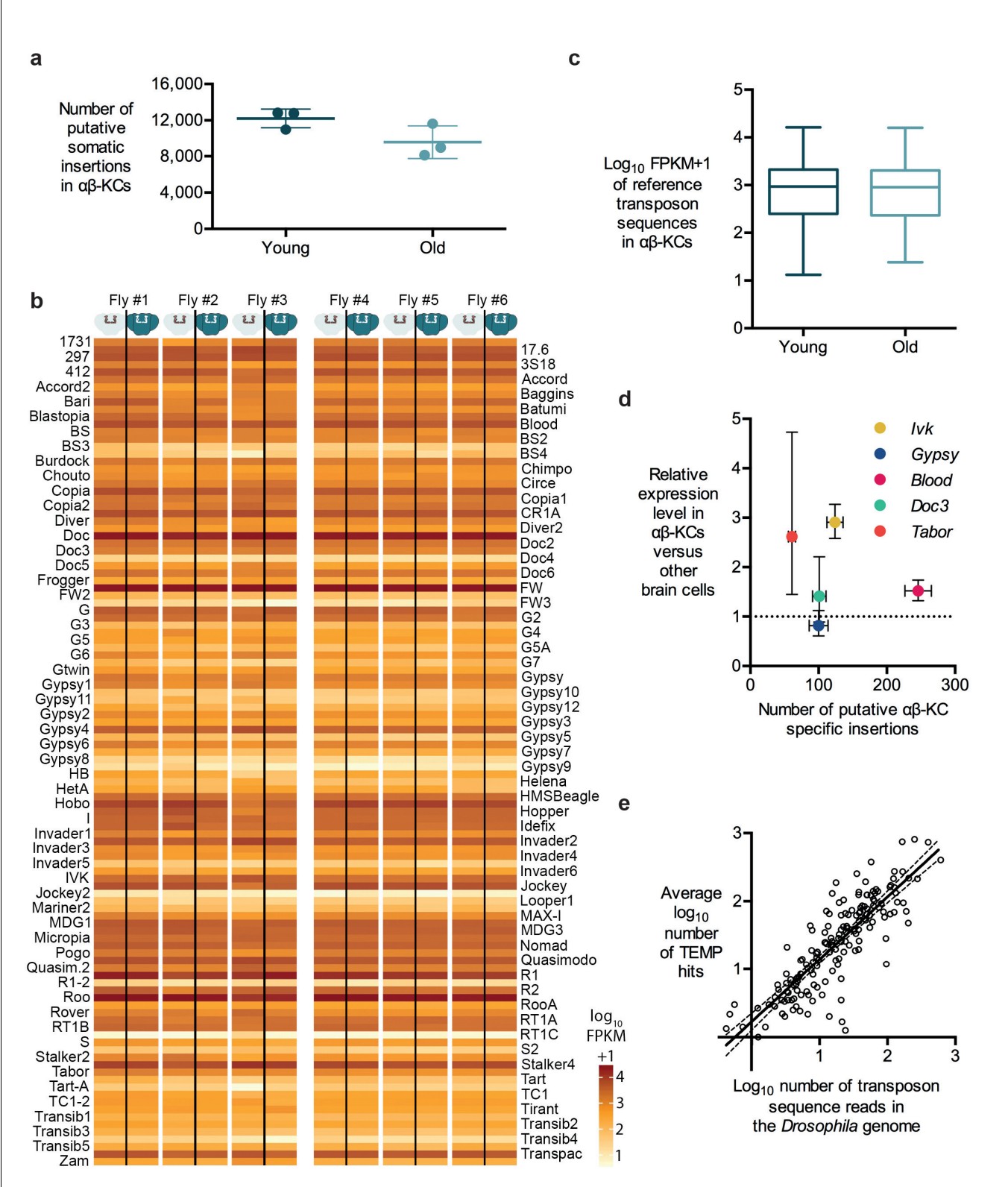

**Figure 3.** Transposon copy number and putative insertion rates do not correlate with age or transposon expression levels. (**a**) The number of putative somatic insertion events does not differ between young (3–4 days) and old (30 days) flies (Mann-Whitney test, p=0.2). Error bars denote SEM. (**b**) Heatmap showing the normalized number of sequencing reads that map onto each of the 111 reference transposon sequences that were analyzed in this study. None of the few visible differences in the amount of transposon sequences in the gDNA from αβ-KCs when compared to the rest of the
*Figure 3 continued on next page*

*Figure 3 continued*

brain of the same individual are statistically significant. Individuals #1 - #3 are young flies (3–4 days) and #4 - #6 are old flies (30 days). FPKM stands for fragments per kilobase of transposon sequence per million fragments mapped. (**c**) Boxplot showing the normalized number of reads that map onto each of the 111 reference transposon sequences per αβ-KC sample of young (3–4 days) and old (30 days) flies. Whiskers represent Min and Max and the box represents the first and third interquartile interval. No statistical difference was evident (Mann-Whitney test, p=0.9184). (**d**) Plot showing no linear correlation between the expression levels of 5 different transposons in αβ-KCs and the number of putative new insertions of each transposon identified in these cells. gDNA data was acquired from 6 independent biological replicates. Error bars denote SEM. (**e**) Plot showing the logarithmic number of reads that map to each transposon consensus sequence taken from the *Drosophila* genome on the x-axis, and the logarithmic average number of putative insertions in 6 flies. Each point represents one transposon type. The line depicts the linear regression ($R^2$ = 0.7166) and the 95% confidence interval.

were more new insertions of the retrotransposon *roo* than the *P* element, *roo* is the most abundant transposon in the reference genome. Random sampling of sequences along the genome, would be expected to sample more abundant transposons more frequently, and therefore result in more incorrectly identified de novo insertions.

## Breakpoint spanning reads reveal amplification artefacts

TEMP analysis calls a new insertion if two paired-end reads map further apart than expected (*Zhuang et al., 2014*). This approach does not usually retrieve the sequence spanning the transposon to chromosome breakpoint, so characteristic features of bona fide retrotransposition, such as polyA tails and target site duplications cannot be used to support identification of genuine rare events. We therefore developed Merged Read Temp (MRTemp), a new method that provides the full DNA sequence of transposon: chromosome and IGE: chromosome breakpoints throughout the genome.

For MRTemp, we repeated the αβ-KC purification from individual flies using FACS, but then generated 350 basepair long gDNA fragments and sequenced 250 nucleotides from each end. Next, all read pairs that overlapped with at least 10 consecutive complementary nucleotides at their ends were merged to form full-length DNA contigs. On average, these contigs were actually 270 basepairs long (*Figure 5a*). These contigs were then used to in silico generate two 100 nucleotide long paired-end reads from fragments that have a precise length of 250 basepairs. To our surprise ~4% of these read pairs mapped further apart than the expected 250 nucleotides when they were aligned to the reference genome (*Figure 5b*).

We next determined whether the transposon-sequence containing MRTemp reads included intact transposon ends, which may be indicative of a *bona-fide* insertion. This analysis revealed that breakpoints were evenly distributed throughout the length of all types of transposons (*Figure 5c*). A lack of preference for breakpoints to occur at transposon ends is more compatible with the occurrence of sequence artefacts than genuine transposon insertions. Since our prior analyses also questioned the prevalence of genuine transposition, we hypothesized that the aberrant mapping of MRTemp reads resulted from the formation of chimeric DNA molecule artefacts that arose during DNA amplification. Chimera have previously been detected in MDA amplified bacterial DNA and a fraction of these were formed by ectopic priming of one replicated DNA strand on a nearby strand (*Lasken and Stockwell, 2007*). This process results in the formation of a complementary sequence at the breakpoint of both fragments and the length of these overlaps represent the balance between the efficiency of the priming and the probability of a complementary sequence occurring close to the original displacement site. For the single bacterial DNA molecule that was analyzed, the distribution of the size of these complementary sequences ranged from 2 to 21 nucleotides, with a peak at 5 basepairs (*Lasken and Stockwell, 2007*). When we plotted the lengths of complementary sequences in chimera from our MRTemp sequencing data, we observed a remarkably similar distribution (*Figure 5d*). Importantly, if these amplification chimera happen to occur in proximity to a germline transposon (or normal location of an IGE), then some read pairs that span the newly formed 'breakpoint' will contain one read that maps to a transposon (or IGE), and the other to a sequence that is located up- or downstream of the original location. Consistent with this model, the alignment of the 250 basepair MRTemp reads to the reference genome revealed a broad distribution of theoretical fragment lengths (see *Figure 5b*). Since all input read pairs were created in silico to be exactly 250

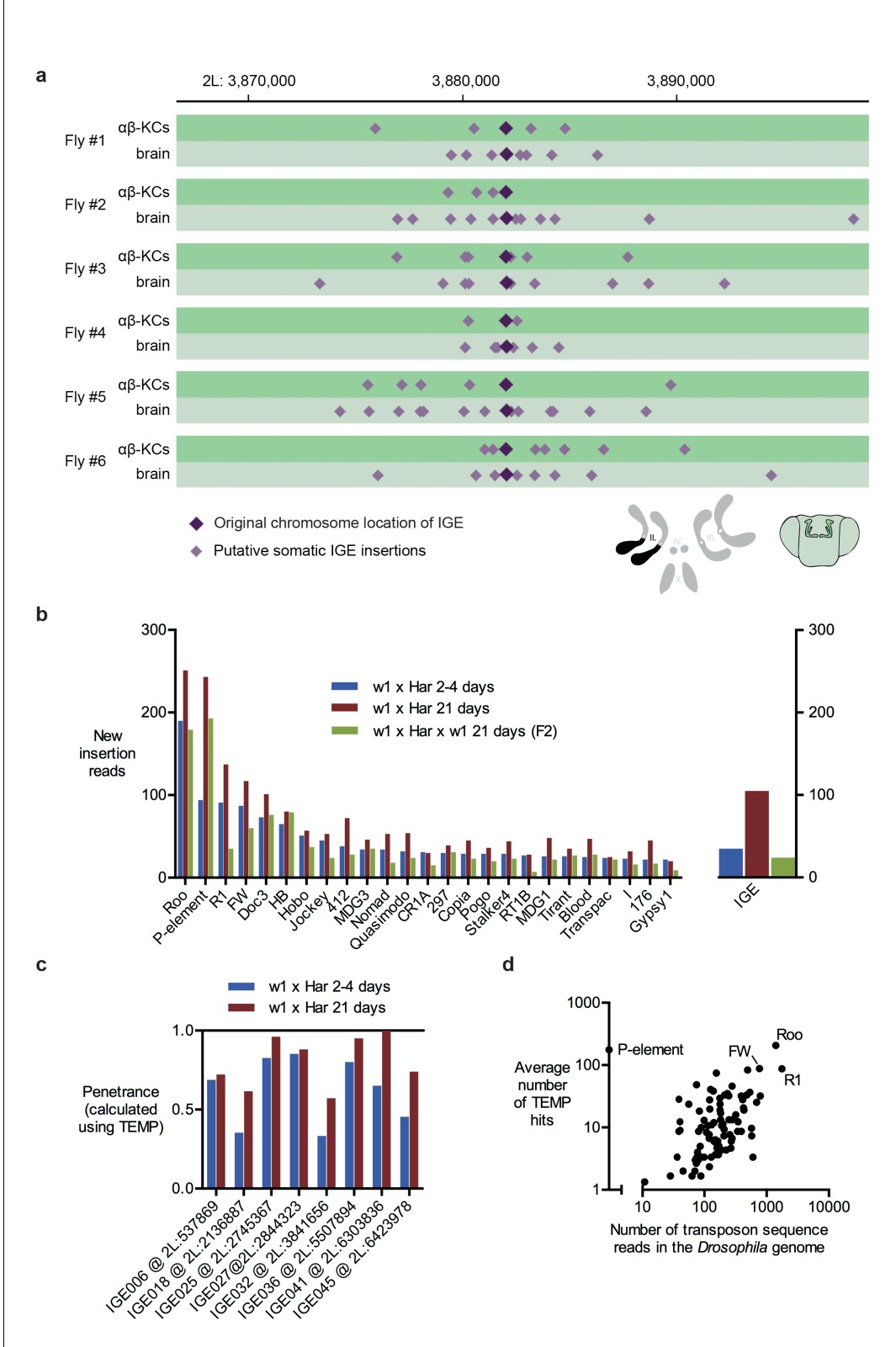

**Figure 4.** Immobile genetic elements appear to mobilize. (a) Representative example of a germline IGE insertion event that was found in each of the 6 individual flies and that is absent in the simulated *Drosophila melanogaster* reference genome (DMsim). Putative somatic insertion events of the same IGE occurred at loci which are approximately 10 kb up- and downstream of the original germline insertion site. Dark purple diamonds represent the original germline insertion site, which, as expected, was present in each of the 12 samples (αβ-KCs and the rest of the brain). Light purple diamonds

*Figure 4 continued on next page*

*Figure 4 continued*

represent putative somatic IGE insertions in each of the samples. (**b**) Plot showing the number of putative new insertions in WGS data from oocytes (*Khurana et al., 2011*). Samples were normalized to a depth of 18.3-fold (as in *Khurana et al., 2011*) and the bars represent the putative insertions that were not detected in the parental strains. In addition, the number of IGE 'mobilizations' is shown. Note that the number of false positive IGE insertions is highest in the sample obtained from 21 day old dysgenic ovaries, and decreases in the F2 generation. (**c**) Graph illustrating the penetrance of a small selection of simulated IGE insertions. The actual penetrance for each locus should be 1. However, due to variations in local sequencing coverage, the analysis pipeline assigns varying frequencies to each IGE insertion. The penetrance of each IGE insertion shown apparently increases with age. (**d**) The average number of putative de novo insertions present in the three samples from oocytes (*Khurana et al., 2011*) correlates with the theoretical number of sequencing reads that map onto each transposon sequence in the *Drosophila* reference genome. The number of reads was based on the sequencing coverage (18.3-fold) and the number of 76nt fragments that overlap with each transposon reference sequence. Note, for example, *Roo*, *R1* and *FW* the three endogenous transposons that contribute most frequently to 'insertions' are also the most abundant elements in the reference genome.

basepairs long, this distribution reflects the actual distances in the genome between the two mapped fragments that constitute the chimera. Most chimera occured between fragments that are less than 10,000 nucleotides apart. Strikingly, our previously observed local accumulation, or hovering, of putative transposon (and IGE) insertions around germline locations, occurred across a similar range of distances, suggesting that in vitro DNA amplification chimera account for the majority of observed transposon mobilizations (*Figure 5e*). Somewhat disturbingly, the distance between MRTemp in silico read pairs ranged from several nucleotides up to the whole length of the chromosome. Therefore, it is inherently impossible, with TEMP and most other currently used transposon detection methods, to distinguish genuine somatic transposon insertions that occurred in a cell from chimeric DNA fragments that were formed during the amplification of gDNA. These artefacts therefore currently present an unavoidable obstacle for the identification of rare somatic transposition events.

## Discussion

Prior studies in mammals and flies have documented elevated expression of transposable elements in the brain and have suggested that transposon mobilization could introduce genomic heterogeneity in neurons, in addition to possible deleterious consequences as animals age (*Baillie et al., 2011*; *Krug et al., 2017*; *Li et al., 2013*; *Ostertag et al., 2002*; *Perrat et al., 2013*; *Upton et al., 2015*). In the current study, we again find some transposon-derived mRNAs are relatively more abundant in $\alpha\beta$-KCs of adult *Drosophila* than in other brain cells. It is notable that the transposons we identified in this study are not the same as those in our prior work, which used a different strain to GFP label $\alpha\beta$-KCs (*Perrat et al., 2013*). In addition, neither of our studies indicate that a particular type of transposon is preferentially expressed. In this study, the LINE-like *Ivk* was upregulated in $\alpha\beta$-KCs but *Doc3* was unchanged, whereas the LTR elements *Tabor* and *Blood* were elevated but *Gypsy* was not. At this point it is unclear why certain transposons are more highly expressed in $\alpha\beta$-KCs. One explanation could be that they reside in neural genes, and therefore adopt the specific expression pattern of these genes. Higher levels of transposon-derived RNAs are also consistent with the prior suggestion of differential silencing by PIWI-associated RNAs in neural subtypes (*Perrat et al., 2013*).

To test whether increased transposon expression levels translate to new insertions that impact the integrity of neural genomes, we developed a new approach that permits sequence analysis of gDNA extracted from neurons purified from individual flies. To our knowledge, no other published study has analyzed gDNA from single flies. The single fly approach in contrast to previous *Drosophila* studies, allowed us to unambiguously identify germline transposon insertions that are present at low frequencies in a population of flies. Despite extensive inbreeding, we found that our fly strain contained many rare germline transposon insertions, which were found in both samples from the same fly but were absent in the WGS samples from all other individual flies (see WGS data in methods). Locating new germline insertions demonstrates the high sensitivity of the single fly approach and also illustrates a critical impediment to analyzing rare transposition using gDNA isolated from groups of flies. One such rare germline insertion in the population could very easily be mistaken for a rare de novo somatic insertion.

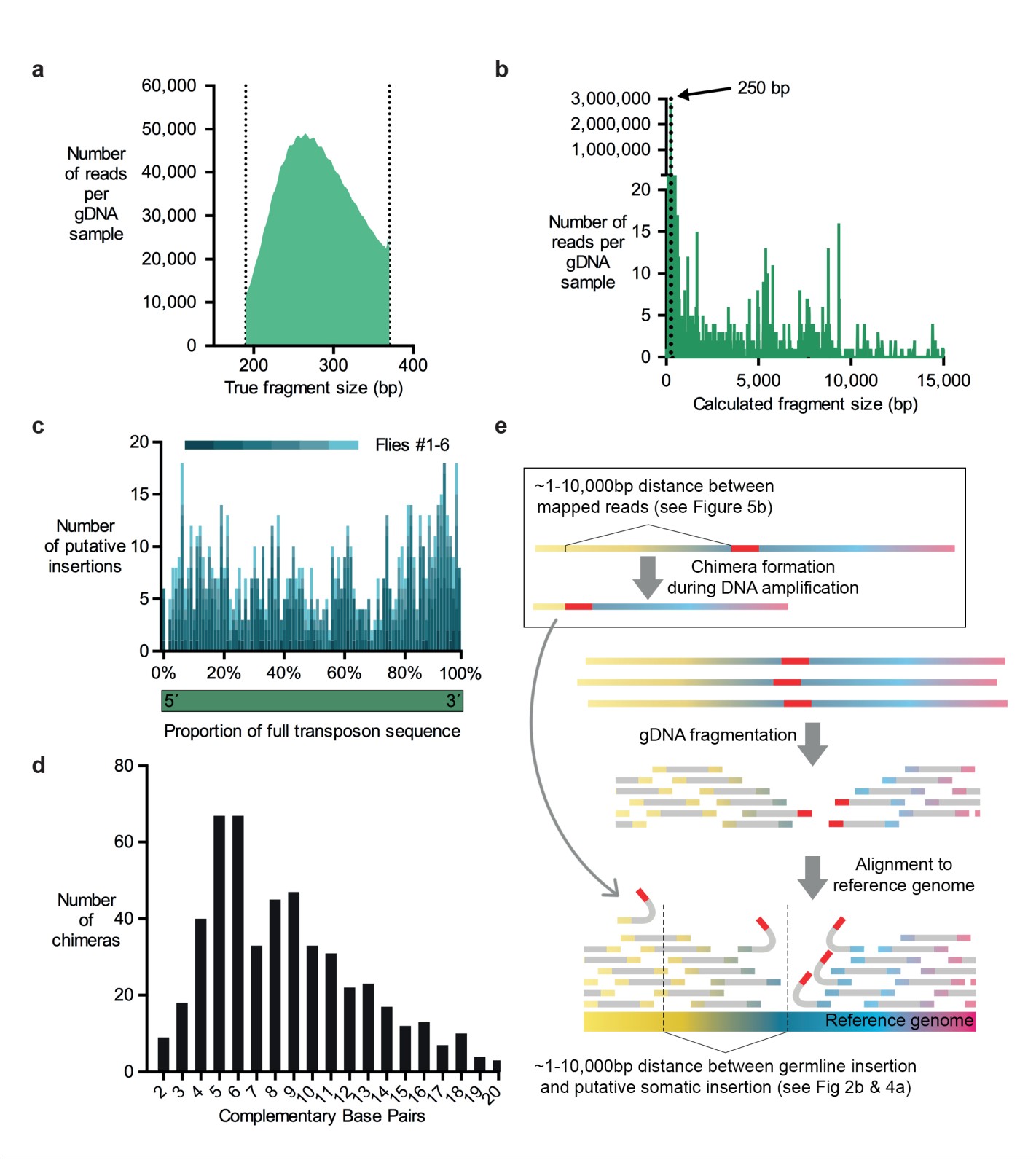

**Figure 5.** Evidence for chimera formation during DNA amplification. (**a**) The DNA fragment sizes of long, overlapping sequencing read pairs were assessed by merging each read pair. The length of these merged fragments varied and peaked at 270 bp. (**b**) In silico assembled read pairs which are all 250 basepairs apart map at genomic locations which are further apart than the predicted size. Plot shows the calculated fragment size which is based on the distance between each of the two mapped paired-end read. (**c**) Transposon: chromosome breakpoints occur across the entire length of

*Figure 5 continued on next page*

*Figure 5 continued*

transposons. Plotted are the number of putative transposon insertions in each fly tested and grouped into bins based on the relative position of the breakpoint along the length of each transposon. (**d**) Graph illustrating the frequency of each size of complementary sequence spanning the junction of chimeric amplicons (only those above 1 are shown). (**e**) Schematic depicting how chimeric DNA, which was formed during gDNA amplification, can result in read-pairs which lead TEMP to predict the presence of a rare somatic transposon insertion down- or upstream of a germline insertion. For TEMP, gDNA is extracted, fragmented, sequenced and paired-end reads are aligned to a reference genome. According to data presented in panel (**b**), during gDNA amplification chimera are favorably formed between sections of gDNA that are between ~1 and 10,000 basepairs apart. This clustering mirrors the range of the apparent transposon and IGE hovering as predicted by TEMP (see *Figures 2b* and *4a*).

Despite the increased transposon expression and the high sensitivity of the mapping approach, we did not detect new transposition events in gDNA from adult fly $\alpha\beta$-KCs beyond the rate of experimental artefacts. These results question the validity of our previous study (*Perrat et al., 2013*) that was performed using gDNA extracted from groups of flies.

Transposons, and the LINE-like *Gypsy* in particular, have also been reported to contribute to age-dependent neuronal decline in flies (*Krug et al., 2017*; *Li et al., 2013*). However, we did not find evidence for transposon insertions accumulating in older flies, even when individual samples were normalized for the estimated rate of chimera formation using our IGE method. We focused our gDNA sequencing experiments on $\alpha\beta$-KCs, a distinct subset of ~1800 neurons in the fly brain that have previously been shown to play a key role in the retrieval of olfactory memory performance and are a plausible neural locus for age-dependent memory loss (*Li et al., 2013*; *Krashes and Waddell, 2008*; *Krashes et al., 2007*). Despite identifying experimental artefacts in every data set, our high sequencing coverage, TEMP and MRTemp analyses, suggest that de novo transposon insertions are not prevalent in $\alpha\beta$-KCs, or other brain cells. We also did not find any evidence to support the model that insertions accumulate with age, in the few old flies that we analyzed. We cannot exclude that in young and/or older flies a small subset of the ~1800 $\alpha\beta$-KCs contain a high number of new insertions, or that many cells contain few insertions. We also cannot rule out that we happened to pick three old, but still healthy flies for our analyses (a similar sampling argument would also apply if we were sequencing single neurons) and/or that the few cells that accumulate transposon insertions in aged flies are removed by cell death and so are not recovered in the material sequenced. However, a reporter for *Gypsy* activity, that we assume is relatively insensitive, was shown to label some $\alpha\beta$-KCs in old flies (*Li et al., 2013*). If the reporter is a genuine measure of *Gypsy* movement, insertions should be prevalent. Moreover, the cells labeled with the reporter are still alive. Since odors are coded as activity in somewhat random sparse populations of $\alpha\beta$-KCs, we also assume that the reported age-dependent decline of odor-specific long-term memory performance (*Li et al., 2013*) would only result if a considerable number of $\alpha\beta$-KCs were damaged. These data are clearly at odds with our complete lack of evidence for transposon mobilization in $\alpha\beta$-KCs, or elsewhere in the brain, where we also did not catch any cells in aged flies that were in the process of accumulating a critical level of disruptive insertions. We therefore propose that the aging phenotype may arise from issues other than transposon-mediated mutagenesis.

To assess the reliability of our gDNA sequencing analysis we developed a new set of WGS control experiments. We used the TEMP method (*Zhuang et al., 2014*) that was established to map new transposon insertions in WGS data to map a collection of exon sequences that we here refer to as immobile genetic elements, or IGEs. Surprisingly, TEMP analysis suggested that IGEs also mobilize in every gDNA sample prepared from mature neurons, suggesting that artefacts are prevalent in WGS data. To further investigate this unexpected finding, we developed MRTemp, which permits the analysis of chromosome rearrangements and transposon insertions and provides nucleotide resolution information of transposon: chromosome and chromosome: chromosome breakpoints. These experiments demonstrated that WGS data are spoiled by chimeric artefacts that are formed by MDA during sample preparation. Since IGE mobilization arises at a similar frequency to that expected of rare somatic transposon insertions, it seems likely that library artefacts account for the majority of instances that have previously been considered to represent rare somatic transposition events.

Applying IGE analysis to *Drosophila* WGS data from another independent study suggests that DNA chimera are a pervasive issue. The rate of mobilization of resident transposons has been

reported to increase with age in the ovaries of P-M hybrid dysgenic female flies (*Khurana et al., 2011*). Our IGE analysis of these data revealed that the apparent increase of somatic transposition events between the two samples could also be explained by an increased abundance of DNA chimera formation in the relevant samples. This difference might have arisen from experimental variations during sample preparation, or from often unavoidable variations in sequencing coverage between different samples. The authors calculated the prevalence, or penetrance, of each insertion by analyzing the number of reads around each putative transposon insertion site. Since penetrance seems to vary randomly for the different elements, it is likely that a subset of insertions can be identified that fits a particular expectation. For example, we were able to identify a subset of IGE insertions, whose penetrance appeared to increase with age in the published data. Importantly, since these IGE insertions are simulated, any apparent increase cannot be real which suggests that the standard TEMP approach is an unreliable indicator of insertion penetrance. IGE analysis therefore identifies a number of crucial caveats of the TEMP approach, which suggests that it will be necessary to re-evaluate prior studies that have employed it to assess somatic transposition.

MRTemp merges two longer overlapping read pairs into a 250 bp fragment, which means that the complete sequence of each stretch of DNA is recovered rather than just the ~100 bp flanking segments that are obtained from the typical paired-end sequencing approach. We used the classic TEMP method to align 100 nucleotides from each end of these merged reads to the *Drosophila* reference genome. As a result, the two MRTemp ends should span exactly 250 nucleotides of the reference genome. The finding that a substantial proportion of read pairs mapped further apart than expected could have at least three reasons. Firstly, they could be genuine transposition events. Secondly, the seemingly longer reads could arise from small deletions that existed in the gDNA of some cells. Thirdly, two DNA fragments could have been merged during the preparation of gDNA for sequencing. Since MRTemp retrieves the sequence across putative breakpoints, we were able to analyze these sequences in detail. Transposon: chromosome fusion showed no preference for transposon ends. We also noted that around 17% of reads that spanned breakpoints contained overlapping palindromic sequences around their breakpoint. Prior work analyzing DNA from *Escherichia coli* suggested that these palindromic sequences are indicative of chimera formed during gDNA library preparation (*Lasken and Stockwell, 2007*). Strikingly, the length of these sequences was remarkably similar in our samples and in the *Escherichia coli* data.

DNA amplification chimera have previously been implicated in the generation of false-positive transposon predictions (*Evrony et al., 2012*, *Evrony et al., 2016*), but to our knowledge we provide the first experimental evidence that MDA produces overlapping complementary sequences when preparing gDNA libraries from eukaryotic material. Demonstrating that DNA chimera present such an impediment to identifying rare transposon insertions is alarming (*Baillie et al., 2011*; *Khurana et al., 2011*; *Evrony et al., 2012*; *Perrat et al., 2013*; *Upton et al., 2015*). Our findings suggest that all gDNA sequencing data from flies, rodents, humans and plants, that were prepared using MDA, are subject to the same amplification artefacts. Moreover, since the chimera can be formed at an early stage of the process, the validation of putative de novo insertions using PCR-based approaches on the same material used for WGS is not an adequate test of the putative insertions being genuine (*Upton et al., 2015*). Our general IGE approach can be adapted to quantify the abundance of artefactual chimera in gDNA data sets prepared from any organism. One only needs to produce a species relevant set of IGEs and a reference genome equivalent to DMsim. It may eventually be possible to develop new amplification strategies so that future sequencing data contain lower levels of amplification chimera. IGE analysis provides a useful method to assess the fidelity of new amplification approaches. An absence of apparent IGE mobilization in sequencing data would indicate a high-fidelity approach and make it possible to identify genuine somatic transposon insertions.

Unfortunately, our analyses lead us to conclude that with current technologies the only convincing evidence for a rare somatic insertion would be breakpoint specific sequence from both ends of a transposon at a specific locus, and in addition, a sufficiently high sequencing coverage from around the same locus in other tissue from the same individual that shows the insertion is not present. The caveats and pitfalls revealed here also emphasize the importance of assessing the prevalence of artefacts of deep sequencing data, especially given the increased use of deep-sequencing in biological research and in medical diagnostics.

# Materials and methods

## Flies

Flies were raised and aged on standard food at 25°C, and 40–50% humidity. Mb008b males were crossed with *w-; +; UAS-mCherry* virgin females.

## Immunostaining

Fly brains for confocal imaging were prepared using a standard immunostaining protocol (*Wu and Luo, 2006*). In summary, brains were dissected in phosphate-buffered saline (PBS, 137 mM NaCl, 2.7 mM KCl, 10 mM Na2HPO4, 1.8 mM KH2PO4) and fixed with freshly prepared 4% Paraformaldehyde in PBT (PBS, containing 0.3% Triton X-100) for 20 min. After fixation, brains were washed $3 \times 20$ min in PBT. Next, brains were incubated in 5% normal goat serum (NGS) in PBT for 30 min, and then incubated with the primary antibody NC82 (Mouse, 1:20 dilution in 5% NGS in PBT) at 4° C for 1 week. After incubation, the brains were washed $3 \times 20$ min in PBT, and then incubated with the secondary antibody (Alexa Fluor 488 Goat Anti-mouse, 1:100) at 4°C for 1 week. Finally, brains were washed $3 \times 20$ min in PBT, and $3 \times 20$ min in PBS, and mounted on microscopy slides (Superfrost PLUS), by carefully placing them between two supporting spacers, and immersing them in antifade mounting media (Vectashield). Brain images were taken on a Leica TCS SP5 confocal LASER microscope, with a HCX IRAPO L $25.0 \times 0.95$ objective. Image stacks were merged and processed with Fiji (*Schindelin et al., 2012*). Only global changes to brightness and contrast were performed.

## Cell preparation

The fly brain dissociation protocol was adapted from *Nagoshi et al. (2010)*. Groups of 50 sex-matched flies (25 females, 25 males) and individual flies were dissected on ice in dissecting saline (9.9 mM HEPES-KOH buffer, 137 mM NaCl, 5.4 mM KCl, 0.17 mM NaH2PO4, 0.22 mM KH2PO4, 3.3 mM glucose, 43.8 mM sucrose, 50 µM d(-)−2-amino-5-phosphonovaleric acid, 20 µM 6,7-dinitroquinoxaline-2,3-dione, 0.1 µM tetrodotoxin), and immediately transferred into Schneider's *Drosophila* medium (Gibco). The brains were washed with dissecting saline, and incubated in l-cystein-activated papain (50 U/ml papain, 1.1 mM EDTA, 0.067 mM 2-mercaptoethanol, 5.5 mM cysteine-HCl) at room temperature for 30 min. The digestion was stopped by the addition of 5 volumes of Schneider's medium, followed by two washes with fresh medium. Samples were triturated by repeatedly pipetting the solution up and down with a flame-rounded P200 pipette tip (pre-wet). Finally, the solution was strained through a 20 µm cell strainer (CellTrics, Partec).

## FACS

Fluorescence Activated Cell Sorting (FACS) was performed on a MoFlo Astrios Cell Sorter, at standard settings, and cells were sorted based on 3 criteria. Firstly, the height of the side scatter (SSC) signal at 488 nm was plotted against the height of the forward scatter (FSC) 488 nm signal, and an area was carefully hand-drawn to select droplets with intact cells and to avoid droplets with cell debris. Secondly, the 488nm-FSC signal was used to exclude droplets that contained more than one cell. And thirdly, the 488 nm-513nm signal (area-log, green fluorescence) was plotted against the 561–579 nm signal (area-log, red fluorescence), to exclude droplets with cells or cell debris that emit fluorescence independent of the excitation wavelength, which would be the case for autofluorescence. mCherry-positive and negative cells were selected and sorted for further processing. Samples were visually inspected in a fluorescence stereomicroscope to confirm the purity.

## RNA isolation

For gene expression assays, mRNA was extracted from FACS fractions with the Arcturus PicoPure RNA isolation kit, using the accompanying protocols. In summary, cells were incubated in extraction buffer at 42°C for 30 min, and cell debris were removed by centrifugation (3000 x g, 2 min). RNA was isolated on RNA purification columns, treated with DNAseI, to remove DNA contamination, and finally eluted with 20 µl elution buffer. Next, RNA was in vitro transcribed into cDNA using the Superscript III First-Strand Synthesis kit (Invitrogen) and oligo-dT$_{(20)}$ primers. For each sample, a control without RT (-RT) was also generated, in order to test for potential gDNA contamination.

### RT-qPCR on FAC-sorted cells

cDNA levels from FAC-sorted cells were analyzed on a Roche LightCycler 480 II, using the Roche Universal ProbeLibrary (UPL) assay. First, primers and probes were generated using the online UPL assay design center. All primer pairs were pre-tested on an RNA dilution series from whole flies, and only primers with linear Ct values were used. cDNA was always amplified in batches of 8 primer pairs (2 primer pairs for housekeeping reference genes, and 6 for target genes). First, all primers were pooled, and template cDNA was amplified by PCR, using the HotStart ReadyMix (Kappa biosystem) PCR mix, and 18 PCR cycles. Next, primers were removed using a PCR purification kit (Qiagen). The sample was then split up into three technical replicates, and 8 separate assays per replicate. Finally, the primer pairs and UPL probes for each target- and reference gene were added, together with the UPL mastermix, and analyzed in the LightCycler. -RT samples were treated the same way, and samples with gDNA contamination were discarded.

### RT-qPCR data analysis

Ct-values, obtained from RT-qPCR assays, were used to test the relative levels of cDNA in each sample, using the 2-$\Delta\Delta$CT method (*Livak and Schmittgen, 2001*; *Nolan et al., 2006*). Results from technical triplicates were pooled and normalized to the geometric mean Ct values of the two housekeeping genes (HG) GAPDH and SdHA ($\Delta$Ct of the target – $\Delta$Ct of HGs) (*Barber et al., 2005*). The average and standard deviation (SD) of $\Delta$Cts from each FACS fraction and each independent biological replicate were calculated. Next, a $\Delta\Delta$Ct value was calculated for each gene or transposon, by subtracting the $\Delta$Ct from the rest of the brain from the $\alpha\beta$-KC $\Delta$Cts. A combined standard error of the mean (SEM) was calculated by first converting the individual SDs from each sample into SEM values, and then by taking the square root of the sum of squared individual SEMs. Next, upper and lower limits were calculated by adding and subtracting the SEM from the average. Finally, the mean, upper and lower limit $\Delta\Delta$Ct were converted into fold changes using the formula: $2^{\wedge}(-1 \times \Delta\Delta Ct)$.

### Genomic DNA extraction and amplification

For our gDNA analysis, cells were dissected and FAC-sorted from single flies. 2 µl of PBS was added to each of the sorted cell populations, if the resulting volumes were less than 1 µl. Next, gDNA was amplified with a Qiagen REPLI-g Midi Kit, using the standard protocol (*Dean et al., 2002*). gDNA was extracted using a protocol which was originally provided by Qiagen online, but which has since become unavailable. The REPLI-g reaction was first equilibrated to room temperature, and 150 µl of 96% ethanol were added. The sample was mixed carefully, and then centrifuged at maximum speed for 2 min. The supernatant was carefully removed, and the pellet was washed with 100 µl of 70% ethanol (avoiding re-suspension of the pellet). The samples were centrifuged again for 2 min, and the supernatant was removed, making sure that no ethanol remained in the tube. After 5 min incubation, the pellet was re-suspended in 50 µl of TE (10 mM Tris, brought to pH 8.0 with HCl, 1 mM EDTA) buffer.

To avoid DNA contamination of the sample, all solutions, tubes, forceps and pipettes were UV-incubated for 10 min before they were used, and appropriate PPE was worn.

### Whole-genome sequencing

WGS of short reads was performed by Beijing Genomics Institute (BGI). 500 bp short-insert PCR-free libraries were constructed from the amplified gDNA, and 100 bp paired-end reads were sequenced on an Illumina HiSeq2000 platform, with a theoretical coverage of 30x. WGS of long reads was performed by Macrogen, South Korea. TruSeq DNA PCR-free libraries with 350 bp fragment sizes were prepared, and 250 bp paired-end reads were sequenced on an Illumina HiSeq2500 platform, with a theoretical coverage of 30x (not taking into account the overlap of the two read pairs).

### Immobile genetic elements

A set of IGEs was assembled to replace the set of transposon sequences. The IGEs were chosen by first putting together a list of 1344 exons from the genes that are most highly expressed in adult *Drosophila melanogaster* testes (from FlyAtlas, microarray, adult Canton-S testis). The exons were ranked according to their size, and the largest 590 exons were chosen. Next, these 590 exons were removed from their original location in the reference genome, and appended to the end of

chromosome 2, using BLAST and standard text editing tools. This new reference genome (DMsim), and the list of 590 exons (the IGEs) was used for the control experiments to assess the prevalence of IGE artefacts. Both files can be downloaded from the public repository that accompanies this manuscript.

## Alignments

Paired-end reads were aligned using both the Burrows-Wheeler Aligner (BWA) (*Li and Durbin, 2009*), and also TopHat (*Trapnell et al., 2012*). The *Drosophila melanogaster* reference genome release 5.57 was downloaded from Flybase (*Flybase Consortium, 1996*). The set of transposon reference sequences was based on the *Drosophila melanogaster* RepBase transposon library (*Jurka, 2000*). The same genome and transposon reference sequences were used throughout this study. A modified reference genome (DMclean) was produced by first masking all sequences that mapped to a portion of the set of transposon reference sequences, with RepeatMasker (version 4.06) (*Bedell et al., 2000*), using the Repbase transposon database. Next, the set of transposon sequences was assembled into a new, theoretical chromosome called 'TE'. Only one copy of each transposon family was used. Finally, the location of each transposon was added to the. gtf files, which contains the locations of all exons.

## TEMP

The TEMP algorithm has previously been described (*Zhuang et al., 2014*). TEMP first identifies all discordant read pairs in a sequence alignment file, and then maps them to a set of reference transposon sequences. The reads that map a section of these reference sequences are considered to be the first piece of evidence for an insertion at this particular location. Next, the putative insertion location is screened for additional reads that support the presence, or absence, of an insertion at this particular location. We used single diagnostic reads as cut off for all the transposon insertion mapping in WGS data from fly brains. The number of putative transposon insertions with more than one read were two orders of magnitude smaller than single read hits, similar to a previous report (*Baillie et al., 2011*) (*Figure 2—figure supplement 1*). We also found comparable differences in the numbers of reads representing IGE artefacts. To identify putative insertions that were only detected in $\alpha\beta$-KCs, and not in other brain cells from the same fly, the predicted insertion sites of the other brain cell sample were flanked with 250 bp on each side (to compensate for inaccuracies of the TEMP approach) and overlapped with the $\alpha\beta$-KCs TEMP results using the bedtools suite (*Quinlan, 2002*).

## MRTemp

In order to create 250 bp fragments in silico, long, overlapping reads were first assembled into contiguous DNA fragments with FLASH (Version 1.2.11) (*Magoč and Salzberg, 2011*). Next, theses contigs were filtered to remove fragments that were smaller than 250 bp (28.8%). Of the remaining reads, two 100 bp sections, with a gap of 50 bp in between, were separated, and merged to form 100 bp paired-end reads of 250 bp inserts. Putative insertions were identified with TEMP and the corresponding merged long reads, which also contained the sequence of the 50 bp gap between the two 100 bp reads (and hence the precise breakpoint) was identified. These contiguous fragments were mapped to the reference genome with BLAST to identify the exact location of the breakpoints in the genome. In addition, the reads were also mapped to the reference transposon/IGE sequences, in order to determine the breakpoints inside the element. Processing of genome sequencing data was partly done using the bedtools suite (*Quinlan, 2002*). The distribution of fragment sizes, based on alignment results, was performed on the Galaxy web platform (version 1.136), using the CollectInsertSizeMetrics function, with the following parameters: MINIMUM_PCT = 0.05, DEVIATIONS = 10. To assess the position of the transposon: chromosome along the consensus sequence of each transposon family, we measured the distance between each breakpoint and the opposite end of the transposon. All 3′ breakpoints are plotted in *Figure 5c*.

## Analysis of TE copy numbers

To assess the copy numbers of reads which map to each transposon, each sample was realigned to the DMclean reference genome with TopHat. The number of reads per gene (and transposon) was assessed with CuffDiff.

## Statistics

We assessed gene expression levels in $\alpha\beta$-KCs by determining the SEM of $\Delta\Delta$CT values obtained from 2 independent biological replicates of groups of 50 flies (with 3 technical replicates per sample). To test whether the number of putative somatic insertions in $\alpha\beta$-KC differed between young and old flies, we compared data from 3 individual biological replicates per age-group. We used the Mann-Witney test to assess whether any apparent differences were statistically significant. The same approach was used to analyze FPKM values between the two age groups.

## Data availability

All WGS data, the set of transposon reference sequences, our modified DMsim genome, the list of 'immobile genetic elements' and the list of transposon sequences used in this study have been deposited in the Dryad Digital Repository (*Treiber and Waddell, 2017*).

## Acknowledgements

We thank G Rubin and Bloomington Stock Center for flies. We are grateful to members of the Waddell group for discussion and comments on the manuscript, to E Harrell for his advice and support with the computational analysis of genome data and to N Rust for his help with FACS. CT was supported by a Wellcome Trust PhD studentship and SW is funded by a Wellcome Trust Principal Research Fellowship in the Basic Biomedical Sciences and the Bettencourt-Schueller Foundation.

## Additional information

### Funding

| Funder | Grant reference number | Author |
| --- | --- | --- |
| Wellcome | 200846/Z/16/Z | Scott Waddell |
| Fondation Bettencourt Schueller | | Scott Waddell |

The funders had no role in study design, data collection and interpretation, or the decision to submit the work for publication.

### Author contributions

CDT, Conceptualization, Resources, Data curation, Formal analysis, Validation, Investigation, Visualization, Methodology, Writing—original draft, Project administration, Writing—review and editing; SW, Conceptualization, Resources, Supervision, Funding acquisition, Validation, Writing—original draft, Project administration, Writing—review and editing

### Author ORCIDs

Christoph D Treiber, http://orcid.org/0000-0002-6994-091X
Scott Waddell, http://orcid.org/0000-0003-4503-6229

## Additional files

### Supplementary files

• Supplementary file 1. List of non-reference transposon insertion detected in all 12 WGS samples.

### Major datasets

The following dataset was generated:

| Author(s) | Year | Dataset title | Dataset URL | Database, license, and accessibility information |
|---|---|---|---|---|
| Treiber CD, Waddell S | 2017 | Data from: Resolving the prevalence of somatic transposition in Drosophila | http://dx.doi.org/10.5061/dryad.fd930 | Available at Dryad Digital Repository under a CC0 Public Domain Dedication |

The following previously published dataset was used:

| Author(s) | Year | Dataset title | Dataset URL | Database, license, and accessibility information |
|---|---|---|---|---|
| Khurana JS, Wang J, Xu J, Koppetsch BS, Thomson TC, Nowosielska A, Li C, Zamore PD, Weng Z, Theurkauf WE | 2011 | Adaptation to transposon invasion in Drosophila melanogaster | https://www.ncbi.nlm.nih.gov/sra/?term=SRP007937 | Publicly available at the NCBI Short Read Archive (accession no: SRP007937) |

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
