## [Decision Letter]

Thank you for submitting your article "Resolving the prevalence of somatic transposition in *Drosophila*" for consideration by *eLife*. Your article has been favorably evaluated by Detlef Weigel (Senior Editor) and four reviewers, one of whom, Jonathan Flint (Reviewer #1), is a member of our Board of Reviewing Editors. The following individuals involved in review of your submission have agreed to reveal their identity: Josh Dubnau (Reviewer #2); Haig Kazazian (Reviewer #3); Christopher A Walsh (Reviewer #4).

The reviewers have discussed the reviews with one another and the Reviewing Editor has drafted this decision to help you prepare a revised submission.

Summary:

Following their report that de novo transposon insertions occurs in αβ neurons and that transposon expression is more abundant in αβ neurons than in neighboring mushroom body neurons, Treiber and Waddell here explain that artefacts are responsible for most, if not all, of apparent novo transposon insertions. The authors provide a detailed description of how they discovered the artefact, provide simulations and experimental support for their interpretation, and conclude that the occurrence of chimeras in DNA prepared for next generation sequencing makes the detection of de novo insertions intractable with the current technology. The realization that reports of transposon insertion rates in various studies might be inflated is an important point for the field. The model that transposon mobilization is important in learning and memory and/or aging in flies is widespread, and this paper would provide an important cautionary challenge to those findings.

Essential revisions:

1) The authors need to be more circumspect in the conclusions and re-write the paper to make the case for how their results place important constraints on the possible impacts on biology. There are two issues that need addressing:

a) Their findings suggest that developmental events are at best rare in bulk tissue. In that case, the most likely way they can impact biology is if there are subsets of cells with high rates. They rule out the possibility that a large fraction of the DNA content in the brain contains de novo inserts of retrotransposons that occurred during development. But they cannot rule out the possibilities that many cells have a low rate or that few cells have a high rate. This is just the limitation that comes from bulk sequencing of many cells. The authors can make an important statement even if they scale back their conclusions so that they are better justified. And this may of course impact similar studies using MDA from rodent and human brain. Again, it does not rule out that there are some cells with high rates or many cells with low rates. But it places important constraints on the biology, and raises a critical technical issue that the field needs to consider

b) For the case of retrotransposition during aging and neurodegeneration, the logic is somewhat different because the sickest cells – the ones with the highest rates of transposition, are the ones that would presumably die and be quickly cleared from the brain. Again, the authors rule out the possibility that a large fraction of cells in a few young and a few older animals have accumulated many de novo inserts each. But the data do not rule out the possibilities that many cells have few inserts or that a few cells have many inserts. The authors need to discuss how their findings constrain the possible models rather than concluding that transposition doesn't occur with age because the rate of transposition is below detection from bulk tissue given how many chimera artifacts are in MDA based libraries

2) More information on the performance of their approach to detect insertions is needed.

a) To reassure readers that they can detect true signal, the authors should take examples of known transposition and show they can detect the events. For instance, they could use data from the Khurana et al. (2011) paper. In addition there are two Arabidopsis data sets where bona fide de novo mobilization was detected https://www.ncbi.nlm.nih.gov/pubmed/19734882
https://www.ncbi.nlm.nih.gov/pubmed/19734880

b) They need to supply quality control metrics to measure the number of chimeras in the sequencing data. For this they could use their analyses of the IGE simulations. They should be able to find a relationship between IGE frequency and the false positive rate (furthermore, they might be able to use an IGE mobilization frequency to optimize their calling methods and parameters to account for potentially different false positive rates caused by the different numbers of chimeras in different libraries).

c) They should examine the number of candidate insertions as a function of signal/confidence (e.g., the number of reads supporting an insertion). Given the limited sequencing depth and high candidate insertion counts, it seems that the authors have used a single read support as a cutoff for predicting insertions (see point X – need to provide a more detailed Methods section). The number of insertions would likely dramatically decrease when more supporting reads are required for prediction. For example, Figure 1 in Baillie et al. (Nature, 2011) shows that the number of insertions decreased by two orders of magnitude when two reads were required instead of one, and that with more than two reads, the number of insertions remained relatively stable. The authors should use two, or ideally more reads, and if they do not find enough insertions (at least ~50 insertions), they should increase sequencing depth. They should then validate a subset of candidates with 3' junction PCR and full-length PCR and estimate false positive rates. This would require a careful examination of the validated transposon and junction sequences to rule out chimeras as much as possible. The authors should also correlate insertion counts adjusted for false positive rates with age. The authors need to clearly acknowledge the limitations of their experimental design because bulk sequencing with limited sequencing depth limits the ability to detect low-level mosaic insertions.

3) To improve clarity of their argument, they should present evidence of artifacts earlier in the paper. Several papers have already shown chimeras occurring during preparation of sequencing libraries or genome amplification causing false-positive transposon predictions. The authors only cite Faulkner's papers (Baillie et al., Nature, 2011, Upton et al., Cell, 2015) and Evrony et al. (*eLife*, 2016) to support the proposition that "rates of somatic transposition in mammals are hotly debated," but they fail to acknowledge that the debate centered around the formation and (mis)interpretation of chimeras, which was what caused the different claimed rates of somatic transposition in the three cited papers. They then introduce chimeras as though they were the first to raise the issue of chimeras. They need to revise the Introduction to be more precise regarding relevant previous literature.

It is important to be aware of chimeras appearing as somatic transposon insertions in next-generation sequencing, which this work highlights in its analysis of "immobile genetic elements." The presence of such chimeras has been presented in prior studies (e.g., Evrony et al. Cell, 2012; Neuron, 2015; *eLife* 2016; Erwin et al., Nature Neuroscience 2016). Other than confirming the presence of chimeras, this work does not provide further understanding about how to properly deal with the chimeras to better identify true somatic transposon insertions.

4) They should try alternative TE detection methods for more accurate insertion breakpoints with junction sequences and mechanistic signatures such as target site duplication and polyA tails. Although chimeras cannot be completely ruled out without full-length validation, as demonstrated by Evrony et al. (*eLife*, 2016), they can still improve the quality of their insertion predictions. To accomplish this, they would need to adapt existing methods to work for fly data.

---

## [Author Response]

Essential revisions:

1) The authors need to be more circumspect in the conclusions and re-write the paper to make the case for how their results place important constraints on the possible impacts on biology. There are two issues that need addressing:

a) Their findings suggest that developmental events are at best rare in bulk tissue. In that case, the most likely way they can impact biology is if there are subsets of cells with high rates. They rule out the possibility that a large fraction of the DNA content in the brain contains de novo inserts of retrotransposons that occurred during development. But they cannot rule out the possibilities that many cells have a low rate or that few cells have a high rate. This is just the limitation that comes from bulk sequencing of many cells. The authors can make an important statement even if they scale back their conclusions so that they are better justified. And this may of course impact similar studies using MDA from rodent and human brain. Again, it does not rule out that there are some cells with high rates or many cells with low rates. But it places important constraints on the biology, and raises a critical technical issue that the field needs to consider

b) For the case of retrotransposition during aging and neurodegeneration, the logic is somewhat different because the sickest cells – the ones with the highest rates of transposition, are the ones that would presumably die and be quickly cleared from the brain. Again, the authors rule out the possibility that a large fraction of cells in a few young and a few older animals have accumulated many de novo inserts each. But the data do not rule out the possibilities that many cells have few inserts or that a few cells have many inserts. The authors need to discuss how their findings constrain the possible models rather than concluding that transposition doesn't occur with age because the rate of transposition is below detection from bulk tissue given how many chimera artifacts are in MDA based libraries

We have paid close attention to this advice. We have now provided a careful and measured discussion of the available data from our experiments and how they relate to previous results and ideas. We have edited the section of the Discussion as follows: “However, we did not find evidence for transposon insertions accumulating in older flies, even when individual samples were normalized for the estimated rate of chimera formation using our IGE method. […] We therefore propose that the aging phenotype may arise from issues other than transposon-mediated mutagenesis.”

We have edited the sentence: “Surprisingly, TEMP analysis suggested that IGEs also mobilize in every gDNA sample prepared from mature neurons, suggesting that artefacts are prevalent in WGS data.”.

2) More information on the performance of their approach to detect insertions is needed.

a) To reassure readers that they can detect true signal, the authors should take examples of known transposition and show they can detect the events. For instance, they could use data from the Khurana et al. (2011) paper. In addition there are two Arabidopsis data sets where bona fide de novo mobilization was detected https://www.ncbi.nlm.nih.gov/pubmed/19734882https://www.ncbi.nlm.nih.gov/pubmed/19734880

To provide more information on the performance of our approach, we have included a list of 163 previously unmapped (i.e. not in the reference genome) germline transposon insertions that we detected in every single sample from all 12 individual flies used in this study. These new germline insertions have an average of 785 diagnostic reads per insertion (αβ-KCs and other brain cells of each fly, 24 samples in total) (see [Supplementary-material SD1-data]). This finding is now fully described in the manuscript:

“Since we used TEMP in our previous study (Perrat et al., 2013), we used it again here. […] Importantly, these same 163 insertions were common to all 12 samples prepared from six individual flies demonstrating that they represent germline insertions that are unique to our fly strain and that they have not been mapped before ([Supplementary-material SD1-data]).” We added a figure legend for [Supplementary-material SD1-data]: “List of non-reference transposon insertion detected in all 12 WGS samples.”.

In the Discussion: “To our knowledge, no other published study has analyzed gDNA from single flies. […] One such rare germline insertion in the population could very easily be mistaken for a rare de novo somatic insertion.”

The reviewers requested that we analyse the data from Khurana et al. (2011). We actually did that in the first submission and presented the results in Figure 4. This figure showed the results of our analysis pipeline on raw sequencing data generated by Khurana and colleagues. Importantly, the results match those previously published, suggesting that our analysis is consistent with that of other labs. We have added the sentence: “We first repeated the TEMP analysis on the data published by Khurana et al. (2011) and found very similar rates of apparent transposon mobilization (Figure 4, left panel). However, IGE experiments also detected considerable levels of IGE mobilization (Figure 4, right panel)” to highlight this finding.

The reviewer point above also raises a key issue that we tried, but obviously failed to highlight. A key finding of our study is the conclusion that our reanalysis of the mapping of Khurana et al. (2011) with TEMP and the assessment of the quality of the WGS data with IGEs suggests that the majority, if not all, of the apparently somatic insertions that the authors report, and the differences observed between strains, are likely to result from either rare germline insertions, or are the product of chimera. Although we previously wished to leave this a little vague we have now been more explicit and added these sections to our manuscript: “The different levels of artefacts in each sample therefore follow the apparent observed differences in resident transposon mobilization. […] Although IGE insertions should all have a penetrance of 1 in a perfect WGS data set, we only identified a subset of IGE insertions to have a penetrance of 1 in the Khurana et al. (2011) samples, while others diverge from this value.”

And: “Although Khurana et al. (2011) noted that there were more new insertions of the retrotransposon roo than the P element, roo is the most abundant transposon in the reference genome. Random sampling of sequences along the genome, would be expected to sample more abundant transposons more frequently, and therefore result in more incorrectly identified de novo insertions.”.

We thank the reviewers for bringing the two Arabidopsis studies to our attention. Although we were previously aware of these studies, unfortunately neither study provides access to their original data. Moreover, only a few transposon families exist both in the Arabidopsis and the *Drosophila* genome. Lastly, for the record, the study by Mirouze and colleagues used a very similar approach to Khurana et al. (2011) that was based on paired-end reads to detect somatic transposon insertions, with a cut-off for a positive hit being set at 14 diagnostic reads. However, this high cut-off does not rule out the possibility that chimera were formed during the amplification of DNA and falsely detected as apparent transposon insertions. These stringent parameters actually increase the chance that a genuine germline transposon insertion is only detected in one sample, and hence misinterpreted as a somatic event.

*b) They need to supply quality control metrics to measure the number of chimeras in the sequencing data. For this they could use their analyses of the IGE simulations. They should be able to find a relationship between IGE frequency and the false positive rate (furthermore, they might be able to use an IGE mobilization frequency to optimize their calling methods and parameters to account for potentially different false positive rates caused by the different numbers of chimeras in different libraries).*

We are grateful to the reviewers for their appreciation of the importance of quality control metrics. However, again we failed dreadfully in describing data that were already in the manuscript. We developed the IGE approach for exactly this reason because it allows us to test any WGS data set in a completely unbiased way for the presence of amplification chimera. In the first submission we ran IGE analyses on all our new data and on all previously published data from Khurana et al. (2011). This was presented but we have now highlighted this contribution in our manuscript. In the Introduction: “Although previous studies of somatic transposition raised the issue of chimeric DNA sequences influencing the reliability of mapping transposon insertions (e.g. Evrony et al., 2016), we here provide evidence that chimera are prevalent in WGS data prepared from a eukaryote. Moreover, we present a new approach to assess the abundance of these chimera in any WGS data set.”And in Results: “…IGE analyses are an invaluable quality control metric for next generation sequencing data because the total number of additional IGE insertions is indicative of the false discovery rate for each gDNA sample.”

We have also extended the legend to Table 1, which includes the quality control metrics for each sample that was used in this study: “Note that the number of IGE insertions (column “Putative IGE insertions”) is a useful quality control metric to estimate the rate of chimera formed during amplification. Furthermore, the number of correctly identified IGEs (last column), in combination with the mean sequencing coverage, can be used to assess how equally distributed the read pairs are for each sample.”

*c) They should examine the number of candidate insertions as a function of signal/confidence (e.g., the number of reads supporting an insertion). Given the limited sequencing depth and high candidate insertion counts, it seems that the authors have used a single read support as a cutoff for predicting insertions (see point X – need to provide a more detailed Methods section). The number of insertions would likely dramatically decrease when more supporting reads are required for prediction. For example, Figure 1 in Baillie et al. (Nature, 2011) shows that the number of insertions decreased by two orders of magnitude when two reads were required instead of one, and that with more than two reads, the number of insertions remained relatively stable. The authors should use two, or ideally more reads, and if they do not find enough insertions (at least ~50 insertions), they should increase sequencing depth. They should then validate a subset of candidates with 3' junction PCR and full-length PCR and estimate false positive rates. This would require a careful examination of the validated transposon and junction sequences to rule out chimeras as much as possible.*

We have expanded the relevant Materials and methods section to describe how putative insertions were identified: “We used single diagnostic reads as cut off for all the transposon insertion mapping in WGS data from fly brains. […] To identify putative insertions that were only detected in αβ-KCs, and not in other brain cells from the same fly, the predicted insertion sites of the other brain cell sample were flanked with 250bp on each side (to compensate for inaccuracies of the TEMP approach) and overlapped with the αβ-KCs TEMP results using the bedtools suite (Quinlan, 2002).” We have also added a figure legend for Figure 2—figure supplement 1: “Total number of putative non-reference somatic TE insertions from 12 samples with 1, 2, 3-9 and more than 10 diagnostic reads.”

As the reviewers suspected, the number of putative insertions does indeed decrease when more diagnostic reads are required (although it remained above 50). We have added a new Figure 2—figure supplement 1 with the same metrics which Baillie and colleagues showed in the figure mentioned by the reviewers. We essentially found the same thing. However, it is very important to emphasize that we see the exact same effect for our IGE data so simply increasing the number of reads required does not eliminate the issue of inappropriately attributing an artefact as a genuine rare insertion event. We think another important point to mention is that imposing a higher read count cut-off to identify putative de-novo insertions actually increases the likelihood that a germline insertion that is only identified in one sample, is mistaken for a rare somatic insertion event.

Regarding the suggestion to validate candidates, again the reviewers appeared to have missed an important section of the manuscript. Our new merged read TEMP (MRTemp) approach assembles transposon: chromosome contigs and therefore retrieved breakpoint sequence information for all putative transposon and IGE insertions. We have now included a new figure to illustrate the analysis of breakpoint information where we find that there is no preference for intact ends of the transposons but rather breakpoints appear to occur quite evenly throughout the elements (Figure 5). We believe this indicates that most of these events are unlikely to be genuine. We believe the MRTemp approach is a better thing to do than PCR validation of a handful of putative insertions which could simply validate a few of the artefacts. The MRTemp approach will be discussed again below in point 4.

Figure 5 also has a figure legend: “C Transposon: chromosome breakpoints occur across the entire length of transposons. Plotted are the number of putative transposon insertions in each fly tested and grouped into bins based on the relative position of the breakpoint along the length of each transposon.”

Figure 5 is accompanied by an additional explanation in the Materials and methods section: “To assess the position of the transposon: chromosome along the consensus sequence of each transposon family, we measured the distance between each breakpoint and the opposite end of the transposon. All 3´ breakpoints are plotted in Figure 5.”.

The authors should also correlate insertion counts adjusted for false positive rates with age.

We are grateful for this suggestion and have now done so, using the data available in Table 1. We did not find any correlation with the age of flies using the adjusted insertion counts and have appended the relevant sentence: “However, we did not find evidence for transposon insertions accumulating in older flies, even when individual samples were normalized for the estimated rate of chimera formation using our IGE method.”

The authors need to clearly acknowledge the limitations of their experimental design because bulk sequencing with limited sequencing depth limits the ability to detect low-level mosaic insertions.

We have adjusted the manuscript accordingly (see response to Major Revision point 1a-b).

3) To improve clarity of their argument, they should present evidence of artifacts earlier in the paper. Several papers have already shown chimeras occurring during preparation of sequencing libraries or genome amplification causing false-positive transposon predictions. The authors only cite Faulkner's papers (Baillie et al., Nature, 2011, Upton et al., Cell, 2015) and Evrony et al. (eLife, 2016) to support the proposition that "rates of somatic transposition in mammals are hotly debated," but they fail to acknowledge that the debate centered around the formation and (mis)interpretation of chimeras, which was what caused the different claimed rates of somatic transposition in the three cited papers. They then introduce chimeras as though they were the first to raise the issue of chimeras. They need to revise the Introduction to be more precise regarding relevant previous literature.

It is important to be aware of chimeras appearing as somatic transposon insertions in next-generation sequencing, which this work highlights in its analysis of "immobile genetic elements." The presence of such chimeras has been presented in prior studies (e.g., Evrony et al. Cell, 2012; Neuron, 2015; eLife 2016; Erwin et al., Nature Neuroscience 2016). Other than confirming the presence of chimeras, this work does not provide further understanding about how to properly deal with the chimeras to better identify true somatic transposon insertions.

We have added the following sentences to our Introduction: “However, the prevalence of rare somatic transposition is debated due to difficulties in mapping genuine events using whole-genome DNA sequencing (Baillie et al., 2011; Evrony et al., 2012; Evrony et al., 2016; Upton et al., 2015).”

“Although previous studies of somatic transposition raised the issue of chimeric DNA sequences influencing the reliability of mapping transposon insertions (e.g. Evrony et al., 2016), we here provide evidence that chimera are prevalent in WGS data prepared from a eukaryote.”

And this section to the Discussion: “DNA amplification chimera have previously been implicated in the generation of false-positive transposon predictions (Evrony et al., 2012; Evrony et al., 2016), but to our knowledge we provide the first experimental evidence that MDA produces overlapping complementary sequences when preparing gDNA libraries from eukaryotic material. […] Moreover, since the chimera can be formed at an early stage of the process, the validation of putative de novo insertions using PCR-based approaches on the same material used for WGS is not an adequate test of the putative insertions being genuine (Upton et al., 2015).”

We now also offer an idea of how one can use IGE analysis to assess the presence of chimeras in a WGS data set:

“Our general IGE approach can be adapted to quantify the abundance of artefactual chimera in gDNA data sets prepared from any organism. […] An absence of apparent IGE mobilization in sequencing data would indicate a high-fidelity approach and make it possible to identify genuine somatic transposon insertions.”

4) They should try alternative TE detection methods for more accurate insertion breakpoints with junction sequences and mechanistic signatures such as target site duplication and polyA tails. Although chimeras cannot be completely ruled out without full-length validation, as demonstrated by Evrony et al. (eLife, 2016), they can still improve the quality of their insertion predictions. To accomplish this, they would need to adapt existing methods to work for fly data.

As mentioned above, the reviewers appeared to have missed the importance of our new MRTemp approach which retrieves all breakpoint information for both transposon: chromosome and IGE: chromosome. To our knowledge MRTemp is the first transposon insertion detection method that systematically reveals the sequence around every breakpoint. As mentioned above the retrieved breakpoint sequences show no preference for transposon ends and they are instead distributed across the elements. This is alarming and is consistent with most of the putative insertions being artefacts. This result is now illustrated in Figure 5. In addition, MRTemp revealed that a subset of chimera are formed around complementary sequences (see Figure 5), a hallmark of chimera that are formed during MDA amplification.